# A Random Matrix Perspective on Mixtures of Nonlinearities in High Dimensions

## Abstract

One of the distinguishing characteristics of modern deep learning systems is that they typically employ neural network architectures that utilize enormous numbers of parameters, often in the millions and sometimes even in the billions. While this paradigm has inspired significant research on the properties of large networks, relatively little work has been devoted to the fact that these networks are often used to model large complex datasets, which may themselves contain millions or even billions of constraints. In this work, we focus on this high-dimensional regime in which both the dataset size and the number of features tend to infinity. We analyze the performance of a simple regression model trained on the random features $F = f(WX + B)$ for a random weight matrix $W$ and random bias vector $B$, obtaining an exact formula for the asymptotic training error on a noisy autoencoding task. The role of the bias can be understood as parameterizing a distribution over activation functions, and our analysis directly generalizes to such distributions, even those not expressible with a traditional additive bias. Intriguingly, we find that a mixture of nonlinearities can outperform the best single nonlinearity on the noisy autoecndoing task, suggesting that mixtures of nonlinearities might be useful for approximate kernel methods or neural network architecture design.

## 1 Introduction

It is undeniable that in recent years deep learning systems have found widespread success in their applications to a diverse and ever-expanding set of domains. The foundational results on many tasks such as image recognition (Krizhevsky et al., 2012), speech recognition (Hinton et al., 2012), and machine translation (Wu et al., 2016), have begun to make their way into higher-level products that people interact with and rely upon in their daily lives. Whether these products generate a medical diagnosis, a navigation decision, or some other important output, it is crucial to understand the inner-workings of the deep learning algorithms that generate them.

Unfortunately, our theoretical understanding of these deep learning algorithms continues to lag behind their impressive practical successes. One main challenge in building a fuller understanding stems from the fact that deep neural networks are complex nonlinear functions that employ millions or even billions (Shazeer et al., 2017) of parameters. Traditional wisdom would suggest that to this parameter complexity corresponds an optimization difficulty. Recent work, however, suggests that as the width of a network's hidden layers becomes large, the loss function simplifies and a theoretical analysis becomes tractable (Jacot et al., 2018; Mei et al., 2018; Chizat & Bach, 2018a; Mei et al., 2019; Rotskoff et al., 2019; Rotskoff & Vanden-Eijnden, 2018). In some scenarios, the simplification is such that throughout training the parameters of the model stay within an infinitesimal radius of their initial (random) values, implying that much about neural network training can be understood by studying the random initialization (Jacot et al., 2018; Chizat & Bach, 2018b; Lee et al., 2019).

Another main challenge in building a rich understanding of deep learning systems stems from the fact that they are often trained on very large, complex datasets: even if the models themselves are very large, they may not be large in comparison to the number of constraints they are designed to satisfy. Indeed, many important phenomena may become apparent only by examining the high-dimensional regime in which the dataset size and width are both large and of the same order.

In this work, we focus on the high-dimensional regime and analyze the performance of a regression model trained on the random features $F = f(WX + B)$ for a random weight matrix $W$ and random

bias vector $B$. We obtain an exact formula for the training error on a noisy autoencoding task in the limit that the width and dataset size both go to infinity. The result is determined by the resolvent of the kernel matrix $F^t F$, whose properties we analyze via the *resolvent method* from random matrix theory. Our analysis also provides an exact formula for the eigenvalue density of the kernel matrix, which may be of independent interest since it provides a characterization for how spectral properties of the data covariance matrix propagate through neural network layers at initialization.

## 1.1 OUR CONTRIBUTIONS

The main contribution of our work is an exact characterization of the training error of a ridge-regularized random feature regression model on a noisy autoencoder task in the high-dimensional regime. This is one of the first non-trivial models to be solved exactly in the joint limit of large data and large width and provides an interesting testing ground in which to analyze this regime. Some of our additional contributions include,

- An exact characterization of the spectral density of the randm feature matrix $F = f(WX + B)$, extending prior results of Pennington & Worah (2017) to non-Gaussian data distributions and to non-zero bias distributions.

- One interpretation of the random additive bias is that it induces a distribution of activation functions parameterized by B, i.e. $f(Z; B) := f(Z + B)$. Our analysis trivially extends to *any* distribution of activation functions $f(\cdot\,; B)$ parameterized by $B$.

- We show that there exists a non-trivial distribution over activation functions that outperforms the best possible single activation function on a noisy autoencoding task.

- Our method of proof introduces a surrogate "linearization" of $F$, $F^{\text{lin}}$, that possesses the same spectral information as $F$. $F^{\text{lin}}$ and its properties are likely to be of further interest and utility in analyzing neural networks in high dimensions.

## 1.2 RELATED WORK

Neural networks have been studied from the perspective of high-dimensional statistics in a number of recent works. Most prior work has focused on the bias-free case. Pennington & Worah (2017) studied the spectrum of the activation matrix $f(WX)$ for iid Gaussian data and derived an analytic expression for the training error of a ridge-regularized random feature model trained on pure noise. It is natural to consider incorporating biases by appending a constant feature 1 to $X$. Unfortunately, this leads to biases that are the same order as the weights, and so the effect disappears in the large dataset limit. Moreover, this modification on the data violates the assumptions of Pennington & Worah (2017). Hastie et al. (2019) study ridgeless interpolation in high-dimensional interpolation for linear features as well as nonlinear random features of iid Gaussian data. Louart et al. (2018) derived a deterministic equivalent for the resolvent of the kernel matrix $F^t F$ which allowed for a characterization of the asymptotic training and test performance of linear ridge regression of random feature models.

Other work has investigated learning dynamics and generalization in the high-dimensional regime (Liao & Couillet, 2018a; Lampinen & Ganguli, 2018; Advani & Ganguli, 2016; Advani & Saxe, 2017) as well as the spectra of more complicated objects such as the Hessian (Pennington & Bahri, 2017) and Fisher information matrix (Pennington & Worah, 2018). From the mathematical perspective, random matrix theory provides natural tools (Silverstein & Bai, 1995a) for analyzing the behavior of neural networks in the high-dimensional regime. Liao & Couillet (2018b) examined spectra for data drawn from Gaussian mixture models; see also (El Karoui, 2010) on the spectra of random kernel matrices.

## 2 PRELIMINARIES

Consider a dataset $X \in \mathbb{R}^{n_0 \times m}$ and the random feature matrix,

$$F = f(WX; B),$$

generated by a single hidden-layer network with iid Gaussian weights $W \in \mathbb{R}^{n_1 \times n_0}$ ($W_{ak} \sim \mathcal{N}(0, \sigma_W^2/n_0)$), activation function $f$, and biases $B = b\mathbf{1}_m^T \in \mathbb{R}^{n_1 \times m}$ (for $b \in \mathbb{R}^{n_1}$). We regard

the second argument of $f$ as parametrizing (continuously or discretely) an ensemble of activation functions. We refer to $B$ (or $b$) as the *bias*, in reference to the important special case $f(WX + B)$ (additive bias). In general, however, we only assume the parameters $b_a$ are such that the measure $\frac{1}{n_1}\sum_a \delta_{b_a}$ converges in distribution to some limiting distribution $\mu_B$, effecting an arbitrary distribution over activation functions.

We assume that $\mathbb{E}\,|f(N;b)|^k$ for $N \sim \mathcal{N}(0,\sigma)$ is finite for all $1 \leq k \leq 3$, $\sigma > 0$, and $b \in$ support$(\mu_B)$. When $\mu_B$ is a single Dirac mass at location $b_0$, the activation function can be written as $f(WX; B) = f(WX; b_0) = g(WX)$ for some single-argument function $g$ (for an additive bias, $g(WX) = f(WX + b_0)$). When this is the case, we say the model has a *single activation function*, as opposed to a mixture or distribution of activation functions.

The quantity of interest for the investigations below is the kernel matrix $\frac{1}{n_1}F^t F$, and in particular its *resolvent*,

$$G(z) = \left(\tfrac{1}{n_1}F^t F - zI\right)^{-1}. \tag{1}$$

As we review in Sec. 4, the optimal regression coefficients of a linear model on the random features $F$ is a simple function of this resolvent.

The high-dimensional regime that we study is the one in which the dataset size $m$, feature dimensionality $n_0$, and hidden layer width $n_1$ all go to infinity at the same rate. In particular, as is standard in the random matrix literature, we assume that we can parameterize the limit in terms of the dataset size $m$ in such a way that there exist two positive constants,

$$\phi := \lim_{m\to\infty} \frac{n_0(m)}{m} \quad \text{and} \quad \psi := \lim_{m\to\infty} \frac{n_0(m)}{n_1(m)}. \tag{2}$$

Note that the resolvent is a random matrix, but as $m$ grows large, its normalized trace becomes a deterministic quantity. In the limit that $m \to \infty$, this quantity is known as the *Stieltjes transform*[1],

$$m(z) := \lim_{m\to\infty} \tfrac{1}{m}\mathrm{tr}\,G(z). \tag{3}$$

Together with an auxiliary transform $\tilde{m}(z)$, defined below, these deterministic quantities completely characterize the asymptotic training error of kernel ridge regression on a noisy autoencoder task in this high-dimensional regime.

The Stieltjes transform frequently arises in random matrix methods as a way to encode the spectra of matrices. In particular, if $\lambda_i$ are the eigenvalues of $\frac{1}{n_1}F^t F$ and the empirical distribution of eigenvalues converges in distribution to some deterministic limiting density as $m \to \infty$,

$$\frac{1}{m}\sum_{i=1}^m \delta_{\lambda_i} \to \mu(\lambda)d\lambda, \tag{4}$$

then (with appropriate technical assumptions), the limiting spectral density itself can be recovered from the Stieltjes transform $m(z)$ via the inversion formula,

$$\mu(\lambda) = \lim_{\epsilon\to 0+} \frac{m(\lambda - i\epsilon) - m(\lambda + i\epsilon)}{2\pi i \epsilon}. \tag{5}$$

The Stieltjes transform then substitutes convergence in distribution for pointwise convergence for all $z$ such that $\Im z > 0$.

## 2.1 Methods for computing the Stieltjes transform

We briefly review two standard methods for computing the Stieltjes transform $m(z)$, the *resolvent method* and the *moments method*.

The resolvent method is an approach for computing the Stieltes transform based on the application of the Schur complement formula to the resolvent itself (or to a closely-related block matrix). Intuitively, as the matrix size becomes large, the minors of the matrix are similar in distribution to the larger matrix, and, moreover, the Cauchy interlacing theorem guarantees that their Stieltjes transforms are

---

[1]The distinction between the Stieltjes transform $m(z)$ and the dataset size $m$ should be clear from context.

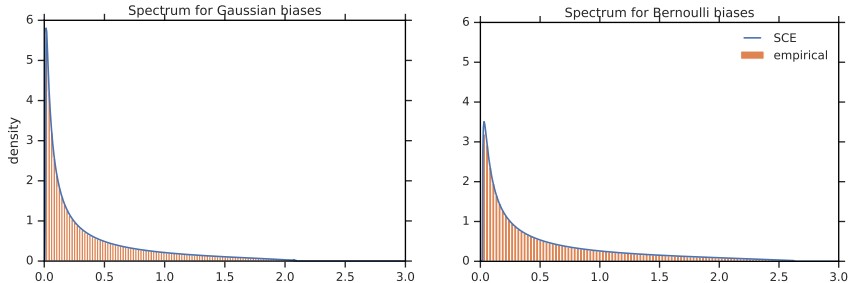

Figure 1: We get excellent agreement of theory and simulation for spectral densities for any bias distribution. We set $\phi = 1.5$, $\psi = 0.8$, $\sigma_X = \sigma_W = 1$, and $f = \text{ReLU}$. Simulations are performed on matrices of size $m = 2^{14}$. **Left** Gaussian distribution over the biases with distribution $\mathcal{N}(0, 1)$. **Right** Bernoulli distribution over biases with distribution $p = 0.5$.

close as well. This allows for the derivation of a self-consistent equation (SCE) in which the Stieltjes transform appears on the left-hand side as the trace of the resolvent, and on the right-hand side as the trace of one of its minors.

The moments method is more combinatorial in nature and involves expanding the resolvent for large $z$ and computing the traces of each term,

$$m(z) = \lim_{m \to \infty} \tfrac{1}{m} \operatorname{tr} G(z) = - \lim_{m \to \infty} \sum_{k=0}^{\infty} \frac{1}{n_1^k} \frac{\operatorname{tr}(F^t F)^k}{z^{k+1}} \ . \tag{6}$$

The traces themselves are expended out as

$$\operatorname{tr}(F^t F)^k = \sum F_{a_1 \alpha_1} F_{a_1 \alpha_2} \cdots F_{a_k \alpha_1} \ , \tag{7}$$

where the sum runs over matrix indices $a_1, \ldots, a_k, \alpha_1, \ldots \alpha_k$. The essence of the moment method involves analyzing the asymptotic contribution of each term in the sum based on its combinatorial type and the details of $F$, and resumming the results to obtain $m(z)$.

We refer the reader to (Erdos & Yau, 2017; Tao, 2012) for more details about these methods and additional background on random matrix theory.

## 3 RESULT FOR STIELTJES TRANSFORM

### 3.1 MAIN THEOREM

We make the following assumptions on the data matrix $X$ and bias vector $b$:

1. $\frac{1}{n_0} \sum_a X_{a\alpha} X_{a\beta} = \delta_{\alpha\beta} \sigma_X^2 + \mathcal{O}\left(1/\sqrt{n_0}\right)$ for all $\alpha$ and $\beta$

2. the empirical eigenvalue distribution of $\frac{1}{n_0} X^t X$ converges in distribution to a measure $\mu_X$

3. $\frac{1}{n_1} \sum_{a=1}^{n_1} \delta_{b_a} \to \mu_B$ in distribution.

**Theorem 1.** *Define $\sigma_Z = \sigma_W \sigma_X$ and resolvent $G(z) = \left(\frac{1}{n_1} F^t F - zI\right)^{-1}$. Then under the above assumptions and for all $z$ such that $\Im z > 0$, the transforms*

$$\tfrac{1}{m} \operatorname{tr} G(z) \quad \text{and} \quad \tfrac{1}{m} \operatorname{tr}\left(\tfrac{1}{n_0} X^t X \operatorname{tr} G(z)\right) , \tag{8}$$

*converge in probability to the unique solution, $m(z)$ and $\tilde{m}(z)$, of the Eqn. (9) that map $\mathbb{C}^+$ to $\mathbb{C}^+$:*

$$m(z) = \mathbb{E}_{S \sim \mu_X}\left[\frac{1}{C_0(z) + SC_1(z)}\right] \quad \text{and} \quad \tilde{m}(z) = \mathbb{E}_{S \sim \mu_X}\left[\frac{S}{C_0(z) + SC_1(z)}\right] \tag{9}$$

*where,*

$$C_0(z) := -z + \mathbb{E}_{B \sim \mu_B}\left[\frac{\eta(B) - \zeta(B)}{D(B)}\right] , \quad C_1(z) := \mathbb{E}_{B \sim \mu_B}\left[\frac{\zeta(B)}{D(B)}\right] , \tag{10}$$

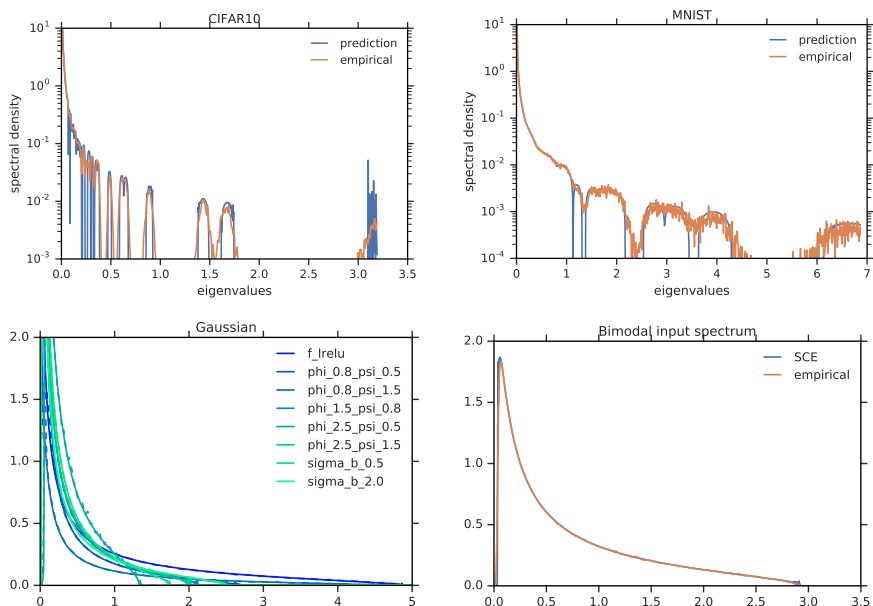

Figure 2: Empirical spectral densities agree with our predictions for varied data distributions and shape parameters. **Top left**: One class from CIFAR (airplane), mean subtracted. **Top right**: Classes $\{0, 8\}$ from MNIST, mean subtracted. **Bottom left**: Gaussian input data, varying the NN parameter settings and activation function. **Bottom right**: Input data with a bimodal spectrum. All plots used $f = \text{ReLU}, \phi = 1.5, \psi = 0.8$ and $\sigma_W = \sigma_B = 1$, except for the indicated modified parameter. Empirical densities were smoothed using a Gaussian KDE.

$$D(B) := 1 + \frac{\psi}{\phi} \left( \zeta(B)\tilde{m}(z) + \left( \eta(B) - \zeta(B) \right) m(z) \right) , \tag{11}$$

*and where $\eta(B)$ and $\zeta(B)$ are the Gaussian expectations*

$$\eta(B) := \mathbb{E}_{N \sim \mathcal{N}(0, \sigma_Z)} \left[ f(N; B)^2 \right] \quad and \quad \zeta(B) := \mathbb{E}_{N \sim \mathcal{N}(0, \sigma_Z)} \left[ \sigma_Z f'(N; B) \right]^2 . \tag{12}$$

The proof is quite involved and is presented in the supplementary material. The basic idea is to derive a multivariate Gaussian random matrix model with the same correlation structure as $F$, then derive a self-consistent equation (SCE) using the resolvent method for this *linearized* version of $F$, $F^{\text{lin}}$.

**Remark 1.** *The self-consistent equations consist of two* coupled *equations involving the Stieltjes transform $m(z)$ and an auxiliary object $\tilde{m}(z)$, (cf. (Paul & Silverstein, 2009, Eq. (2))), which we will see in Cor. 1 essentially measures the autoencoding capacity of the network.*

**Remark 2.** *Note that the self-consistent equations contain an expectation over the limiting spectral density of the input data. While the assumptions on the data matrix $X$ in Thm. 1 are quite general, they may not be optimal. See Sec. 3.3, where we show strong agreement with empirical data from MNIST and CIFAR-10 and for a range of synthetic distributions. This suggests that the theorem may hold for even more general data distributions.*

### 3.2 ALTERNATE REPRESENTATION AND LIMITING RESULTS

When the data distribution is iid Gaussian, the expectations in Eqn. (9) can be expressed in closed form, though one must be careful to choose the correct branch of the resulting function. For simplicity and future reference, we focus on the setting where $0 < \phi \leq \psi \leq 1$, in which case we have the

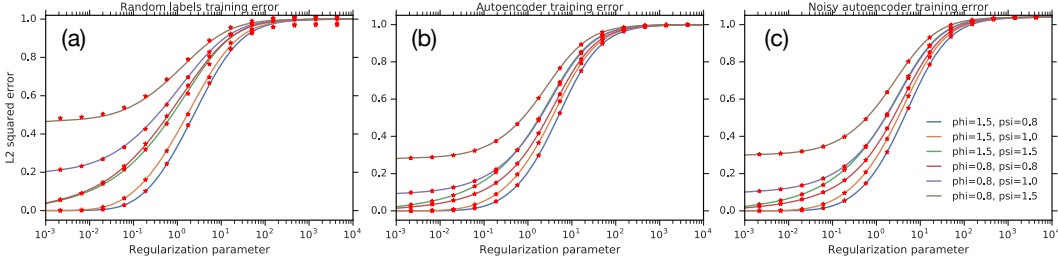

Figure 3: Comparisons of simulated ridge regression error and our theoretical prediction. We use ReLU with $\sigma_X = \sigma_W = \sigma_B = 1$ for all plots and vary the shape parameters $\phi$ and $\psi$. For simulations we use $m = 2^{13}$ throughout. Note we also normalize the activation function so that $\mathbb{E}_b[\eta(b)] = 1$. (a): Our predictions for ridge regression with random labels are solid lines. Simulated losses are the red stars. (b) Autoencoder error. (c) Noisy autoencoder error with $\sigma_\varepsilon = 0.2$.

coupled algebraic equations,

$$m(z) = \frac{C_1 - (C_0 + C_1)\phi + \sqrt{C_1^2 + 2(C_0 - C_1)C_1\phi + (C_0 + C_1)^2\phi^2}}{2C_0C_1} \tag{13}$$

$$\tilde{m}(z) = \frac{1 - C_0 m(z)}{C_1}. \tag{14}$$

When $\mu_B$ in Eqn. (10) is trivial, i.e. a single Dirac mass, then the result should reduce to the single activation function case with $F = f(WX)$, which was studied in (Pennington & Worah, 2017). Indeed, writing $\eta = \mathbb{E}_B[\eta(B)]$ and $\zeta = \mathbb{E}_B[\zeta(b)]$ for such a distribution, and eliminating $\tilde{m}(z)$ from Eqns. (13) and (14), we find that $m(z)$ satisfies the following quartic polynomial:

$$0 = (z^2\zeta^2\psi^2)m(z)^4 + (2z\zeta^2\psi(\psi - \phi))m(z)^3 + (\zeta^2(\psi - \phi)^2 + z\zeta\phi\psi + z\eta\phi^2\psi)m(z)^2$$
$$+ (\zeta\phi(\psi - \phi) + \phi^2(z\phi + \eta(\psi - \phi)))m(z) + \phi^3 \tag{15}$$

which agrees with the result in (Pennington & Worah, 2017) upon identifying $m(z) = -(1 - \phi/\psi)/z - \phi/\psi G(z)$.

### 3.3 SPECTRAL DENSITY ESTIMATES

The self-consistent equations in Thm. 1 can be solved numerically by iterating Eqn. (9) until convergence, using numerical integration. By utilizing the Stieltjes inversion formula, Eqn. (5), we can extract predictions for the limiting eigenvalue density of $\frac{1}{n_1}F^tF$. The results show close agreement with empirical spectral simulations from several interesting practical datasets and synthetic distributions, see Figs. 1 and 2.

## 4 RIDGE-REGULARIZED NOISY AUTOENCODER

We consider the problem of kernel ridge regression with random features given by $F = f(WX; B)$ and noisy regression targets given by $Y = AX + \epsilon$ for some $A \in \mathbb{R}^{n_2 \times n_0}$ and independent Gaussian noise $\epsilon \in \mathbb{R}^{n_2 \times m}$ such that $\epsilon_{ij} \sim \mathcal{N}(0, \sigma_\epsilon^2)$. As is common in the literature of high-dimensional statistics, we assume an isotropic prior on $A$ such that $A^tA \to \frac{n_2}{n_0}\sigma_A^2 I$ as $m, n_0, n_2 \to \infty$. Note that when $\sigma_A = 0$, we have the pure memorization setting studied by Pennington & Worah (2017).

**Corollary 1.** *Let $W_2^*$ be the minimizer for regularized training loss*

$$\mathcal{L}(W_2) = \frac{1}{n_2 m}||Y - W_2 F||^2 + \frac{n_1}{n_2 m}\gamma||W_2||^2, \tag{16}$$

*with random features $F = f(WX; B)$. Then the asymptotic training error converges in probability as*

$$E_{train} = \frac{1}{n_2 m}||Y - W_2^*F||^2 \to \gamma^2 \frac{d}{d\gamma}\big(\sigma_A^2 \tilde{m}(-\gamma) + \sigma_\epsilon^2 \gamma^2 m(-\gamma)\big). \tag{17}$$

In particular, we see that the derivative of the Stieltjes transform $m'(-\gamma)$ measures the capacity to learn noisy labels, whereas $\tilde{m}'(-\gamma)$ measures pure autoencoding capacity. See Fig. 3 for a comparison between these theoretical predictions and simulation.

*Proof.* The optimal weights for the regularized loss are given by

$$W_2^* = \frac{1}{n_1} Y G(\gamma) F^t \quad \text{for} \quad G(\gamma) = (\frac{1}{n_1} F^t F + \gamma I)^{-1},$$

resulting in training error

$$
\begin{aligned}
E_{\text{train}} = \frac{1}{n_2 m} ||Y - W_2^* F||^2 &= \gamma^2 \frac{1}{n_2 m} \text{tr} \left( Y^t Y G(\gamma)^2 \right) \\
&= \gamma^2 \sigma_A^2 \frac{1}{n_0 m} \text{tr} \left( X^t X G(\gamma)^2 \right) + \gamma^2 \sigma_\epsilon^2 \frac{1}{m} \text{tr} \left( G(\gamma)^2 \right) \\
&= \gamma^2 \sigma_A^2 \frac{d}{d\gamma} \left( \tilde{m}(-\gamma) \right) + \gamma^2 \sigma_\epsilon^2 \frac{d}{d\gamma} \left( m(-\gamma) \right) \\
&= -\gamma^2 \left( \sigma_A^2 \tilde{m}'(-\gamma) + \sigma_\epsilon^2 m'(-\gamma) \right). \quad \square
\end{aligned}
$$

**Remark 3.** *As in (Pennington & Worah, 2017), there is a scaling homogeneity in the $E_{train}$: an increase in the regularization constant $\gamma$ can be compensated by a decrease in scale of $W_2$, which, in turn, can be compensated by increasing the scale of $F$, which is equivalent to increasing $\eta(b)$ and $\zeta(b)$. Owing to this homogeneity, we are free to choose a normalization of the activation function for which $\mathbb{E}_{b \sim \mu_B}[\eta(b)] = 1$.*

### 4.1 NONLINEAR MIXTURES CAN OUTPERFORM SINGLE NONLINEARITIES

The bias term in our random feature model can be viewed as one way of defining a distribution over activation functions. The choice of distribution, in general, affects the performance of the model on a given task – as quantified by the expectations in Eqn. (10). A key benefit of our model and analytical approach is that it permits nontrivial distributions of nonlinear activation functions.

In this section, we build on our results for the training loss on noisy autoencoder tasks to examine the benefits of utilizing nonlinear mixtures. The goal here is not to identify "good" mixtures, since this will clearly be a dataset- and architecture-dependent question, nor is it to identify a large performance gap. Instead, we merely seek to demonstrate a proof-of-principle, namely that there exist non-trivial distributions over nonlinearities that can *provably* outperform the best possible single nonlinearity.

For this analysis, we consider the simplest possible nontrivial distribution over activation functions: a Bernoulli mixture of two different functions. To each of these functions we associate two constants, $\eta$ and $\zeta$, which derive from Eqn. (12) but have no $B$-dependence since each function is a *single nonlinearity*. Concretely, let

$$\eta = \mathbb{E}_{N \sim \mathcal{N}(0, \sigma_Z)} \left[ f(N)^2 \right] \quad \text{and} \quad \zeta = \mathbb{E}_{N \sim \mathcal{N}(0, \sigma_Z)} \left[ \sigma_Z f'(N) \right]^2. \tag{18}$$

For the two functions themselves, we utilize (i) a "pure linear" activation function with $\eta = 1$, i.e. the identity function and (ii) a "pure nonlinear" (Hastie et al., 2019) activation function with $\eta = 1$ and $\zeta = 0$. (The particular purely nonlinear function in (ii) is irrelevant, as our theory predicts and our experiments confirm; see Fig. 4(a)). To be precise, we define for $p \sim$ Bernoulli(p),

$$f_p(x) := \begin{cases} x & \text{if } p = 0 \\ g_{\zeta=0}(x) & \text{if } p = 1 \end{cases}, \tag{19}$$

where $g_{\zeta=0}$ is any function with $\eta = 1$ and $\zeta = 0$ (see, e.g., the functions in Fig. (3) of (Pennington & Worah, 2017)).

The task of computing $E_{\text{train}}$ for $f_p$ is cumbersome but purely algebraic. To see how to proceed, notice that the expectations in Eqn. (10) are simple for $f_p$:

$$C_0(z) = -z + \frac{p}{1 + \psi/\phi\, m(z)}, \quad C_1(z) = \frac{1-p}{1 + \psi/\phi\, \tilde{m}(z)}. \tag{20}$$

Plugging these equations into Eqns. (13) and (14), collecting terms and simplifying yields a set of coupled polynomial equations for $m(z)$ and $\tilde{m}(z)$. Taking the total derivative of these equations with respect to z yields two additional equations which can be solved to express $m'(z)$ and $\tilde{m}'(z)$ in terms of $m(z)$ and $\tilde{m}(z)$. Combining these results produces a polynomial system whose solution[2] encodes $E_{\text{train}}$ through Eqn. (17). Fig. 4(a) shows the result of this calculation in solid lines for various values of $\gamma$, while the $1\sigma$ error bars show empirical simulations with finite networks. The red stars in the figure show that for many values of $\gamma$, the optimal mixture percentage is intermediate, i.e. $0 < p < 1$.

---

[2]Special care must be taken in selecting the correct root of this equation, in accordance with the condition that $m(z) \sim -1/z$ for large $|z|$.

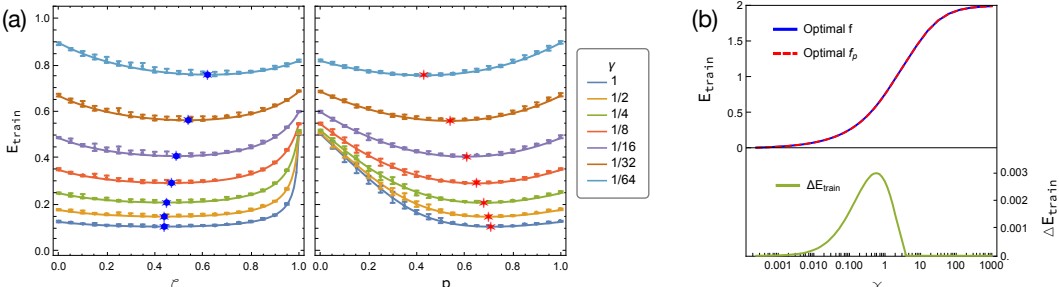

Figure 4: Performance on ridge-regularized noisy autoencoder with $\sigma_\epsilon = 1$, $\phi = 1/2$, and $\psi = 1/2$. (a) Theoretical predictions for training error (solid lines) and $1\sigma$ error bars for empirical simulations of finite networks ($n_0 = 192$, $n_1 = 384$, $m = 384$) for various values of ridge regularization constant $\gamma$ as the activation function varies. In the left panel, a single activation function $f$ is used. In the right panel, the non-linearity is $f_p$, a Bernoulli($p$)-mixture of a purely linear ($\zeta = 1$) and purely non-linear ($\zeta = 0$) function. Each simulation uses a randomly-chosen non-linearity having the specified values of $\zeta$, demonstrating that $E_{\text{train}}$ depends on the non-linearity solely through this constant. Red and blue stars denote minima. (b) Training error as a function of $\gamma$ for the optimal $f$ and $f_p$, as determined in (a). The bottom panel shows the difference in training error, demonstrating that the optimal Bernoulli($p$)-mixture of non-linearities has smaller training error than the best single non-linearity.

The question is, does a non-trivial mixture actually outperform a single nonlinearity? First, we must understand the performance of the optimal single nonlinearity. We note that owing to the homogeneity of the the training loss in $\eta$, $\zeta$, and $\gamma$, we can assume without loss of generality that $\eta = 1$. Therefore the entire effect of the nonlinearity should be encoded in the single constant $\zeta$. In Fig. 4(a), we plot our theoretical prediction for $E_{\text{train}}$ in solid lines and empirical simulations for finite networks as $1\sigma$ error bars. The activation function used for each simulation is chosen randomly, conditional on the value of $\zeta$. So the good agreement in the left panel of Fig. 4(a) demonstrates not just the correctness of our theoretical result but also the fact that $E_{\text{train}}$ depends on the activation function solely through the constant $\zeta$. The blue stars in this figure indicate that the optimal single nonlinearity is neither purely linear ($\zeta = 1$), nor purely nonlinear ($\zeta = 0$), but rather something in between.

For this particular problem setup, the performance of the optimal single nonlinearity and the optimal Bernoulli mixture are rather close, as indicated by the top panel of Fig. 4(b). However, owing to our precise analytical formulation, we can evaluate the training loss to high precision and observe that there is indeed a difference in performance between the two models, as shown in the bottom panel of Fig. 4(b). This result establishes that there are some problems for which even the best single nonlinearity is outperformed by a mixture of nonlinearities.

## 5 CONCLUSIONS

In this work we studied the feature matrix $F = f(WX; B)$ where $W$ is a random matrix with iid Gaussian entries. Under mild assumptions on $X$ and $B$, we obtained an exact analytic formula, Eqn. (9), that characterizes the Stieltjes transform of the spectral density of $F$. The result allowed us to describe the exact training loss of a ridge-regularized noisy autoencoder in the high-dimensional, large-dataset limit, providing one of the first closed-form solutions to a non-trivial model in this limit. We found excellent agreement between the asymptotic predictions of Eqn. (9) and a variety of finite-dimensional empirical simulations.

We also advanced the interpretation of the bias $B$ as one particular way of parameterizing a distribution of activation functions. Indeed, our derivations proceed completely unchanged whether this distribution is of the traditional additive form $f(\cdot + B)$ or the more general $f(\cdot; B)$. By examining the latter, we showed that there are configurations in which a non-trivial distribution over activation functions provably outperforms the best possible single activation function. This opens the door to future investigations regarding optimal methods for parameterizing distributions over activation functions for approximate kernel methods, and suggests the possibility that mixtures of nonlinearities could be a useful design consideration when constructing neural network architectures.

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

# Supplemental Material:
## A Random Matrix Perspective on Mixtures of Nonlinearities in High Dimensions

The main goal of the supplementary material is to prove Thm. 1. This is achieved in three steps. The first step is to note that the limiting Stieltjes for the kernel matrix $F^t F$ is equivalent to a multivariate a multivariate Gaussian random matrix model with an identical correlation structure. This was proven in great generality in Banna et al. (2015) (see Theorem 5.). Similar results for covariance matrices have been previously studied Bai & Zhou (2008), but since $F$ does not have identically distributed rows (or columns) they are inadequate for our case. Specifically, Theorem 5 from Banna et al. (2015), implies that the two Stieltjes transforms have the same limit for all $z$ such that $\Im z > 0$ almost surely.

The second step is to derive the leading order correlation structure (or kernel) for $F$ (Lem. 1), and specify a multivariate Gaussian model with the same structure (Subsec. A.1). We refer to this Gaussian model as a *linearized model*, since it removes the nonlinearity $f$ from the random part of our random matrix ensemble. Without loss of generality, we may assume that $\mathbb{E}F_{a\alpha} = 0$ for all $a$ and $\alpha$. This is because $\mathbb{E}F_{a\alpha}$ only depends on $a$ (as we demonstrate in Sec. A), so we can center $F$ using a rank-1 matrix—a transform that has no effect on the limiting Stieltjes transform as it represents an $\mathcal{O}(1/m)$ change in the spectrum (see for more detail Bai & Zhou (2008)).

The third a final step of the proof is a derivation of a self-consistent equation specifying the limiting Stieltjes transform of the Gaussian model. We do this using the resolvent method (see Erdos & Yau (2017) for an introduction). While the application of this method to multivariate Gaussian covariance matrices is standard, we note the form of the SCE is new and that we include the derivation here for completeness.

At the end of the supplementary material, we include some additional figures.

## A  CORRELATION STRUCTURE OF $F$

To begin we derive an asymptotic form for correlation structure of $F$. For simplicity of presentation, we derive the results for $f(\cdot, B) \equiv f(\cdot + B)$, but the argument generalizes directly. We recall that $W$ is an $n_1 \times n_0$ random matrix with mean 0 and variance $\sigma_w^2/n_0$ iid Gaussian entries. Define $c_{\alpha\beta} := \sum_k X_{k\alpha} X_{k\beta}/n_0$. The data matrix $X$ and bias vector $B$ are deterministic, so all expectations are over $W$ (and can be thought of as conditional on $X$ and $B$). Moreover, $X$ and $B$ satisfy the following assumptions:

- $c_{\alpha\beta} = \delta_{\alpha\beta}\sigma_W^2 + \mathcal{O}\left(1/\sqrt{n_0}\right)$ for all $\alpha$ and $\beta$;
- The empirical spectral distribution of $\frac{1}{n_0}X^t X$ converges in distribution to some measure $\mu_X$;
- $\frac{1}{n_1}\sum_{a=1}^{n_1}\delta_{b_a} \to \mu_B$ in distribution, or equivalently $\frac{1}{n_1}\sum_{a=1}^{n_1}g(b_a) \to \mathbb{E}_{B\sim\mu_B}g(B)$ for bounded, continuous functions $g$.

Observe that

$$Z_{a\alpha} := \sum_{k=1}^{n_0} W_{ak}X_{k\alpha} \text{ and } Z_{b\beta} := \sum_{k=1}^{n_0} W_{bk}X_{k\beta} \tag{21}$$

are jointly Gaussian. Moreover,

$$\mathbb{E}Z_{a\alpha} = 0 \quad \text{and} \quad \mathbb{E}Z_{a\alpha}Z_{b\beta} = \sigma_W^2 c_{\alpha\beta}\mathbf{1}\,(a = b)\,. \tag{22}$$

Note in particular, that $Z_{a\alpha}$ and $Z_{b\beta}$ are independent if $a \neq b$. For convenience, we normalize $c_{\alpha\beta}$ and define $\tilde{c}_{\alpha\beta} := c_{\alpha\beta}/\sigma_X^2$ and $\varepsilon_\alpha := \tilde{c}_{\alpha\alpha} - 1$.

In order to specify the correlations, we employ transforms of the activation function $f$. The transforms are Gaussian integrals of $f$ at different locations and scales:

$$\xi_k(x) := \mathbb{E}\sigma_X^k \sigma_W^k f^{(k)}\left(\sigma_X\sigma_W N + x\right), \tag{23}$$

$$\eta_{i,j}(x) := \mathbb{E}\sigma_X^{i+j}\sigma_W^{i+j}f^{(i)}(\sigma_X\sigma_W N + x)f^{(j)}(\sigma_X\sigma_W N + x), \tag{24}$$

and

$$\gamma_{i,j}(x) := \mathbb{E}\sigma_X^{i+j}\sigma_W^{i+j}f^{(i)}(\sigma_X\sigma_W N + x)f^{(j)}(\sigma_X\sigma_W N' + x) = \xi_i(x)\xi_j(x). \tag{25}$$

**Remark 4.** *For simplicity, while writing $f^{(i)}$ we assume differentiability of $f$ up to $i$th order. However, this assumption can be relaxed, as since the derivatives are only used to take a Gaussian expectation, they can be expressed using Stein's lemma. For example, $\xi_1(x) = \mathbb{E}Nf(\sigma_X\sigma_W N + x)$. Thus, similar results should hold for rectified linear units for example, regardless of their lack of differentiability at 0.*

**Lemma 1.** *To leading order the correlation structure of $F$ is*

$$\mathbb{E}F_{a\alpha}F_{b\beta} = \begin{cases} 0 & \text{if } a \neq b \\ \eta_{00}(b_a) & \text{if } \alpha = \beta \text{ and } a = b \\ \gamma_{11}(b_a)c_{\alpha\beta} & \text{if } \alpha \neq \beta \text{ and } a = b \end{cases}. \tag{26}$$

*Proof of Lem. 1.* Before we prove Lem. 1, we state a simple lemma that is easy to show using Taylor expansion.

**Lemma 2.** *Suppose $\varepsilon$ is a small constant and that $N \sim \mathcal{N}(0,1)$, then*

$$\mathbb{E}_N f\left(\sigma_X\sigma_W\sqrt{1+\varepsilon}N + b\right) = \xi_0(b) + \frac{\xi_2(b)}{2}\varepsilon + \mathcal{O}(\varepsilon^2), \tag{27}$$

$$\mathbb{E}_N f\left(\sigma_X\sigma_W\sqrt{1+\varepsilon}N + b\right)^2 = \eta_{00}(b) + (\eta_{11}(b) + \eta_{02}(b))\varepsilon + \mathcal{O}(\varepsilon^2), \tag{28}$$

*and*

$$\mathbb{E}_N\left(\sigma_X\sigma_W\sqrt{1+\varepsilon}\right)f'\left(\sigma_X\sigma_W\sqrt{1+\varepsilon}N + b\right) = \xi_1(b) + \frac{\xi_1(b) + \xi_3(b)}{2}\varepsilon + \mathcal{O}(\varepsilon^2). \tag{29}$$

Recall $\xi_0(x) = 0$ by assumption. Using the observation in Eqn. (21), we repeatedly rewrite the expectations in $F^tF$ as we can reduce the expectation over $W$ to an expectation over at most two correlated Gaussian random variables. Let $N$ and $N'$ denote independent, standard Gaussian random variables. Using Lem. 2, we see

$$\mathbb{E}f(Z_{a\alpha}) = \mathbb{E}f(\sigma_W\sqrt{c_{\alpha\alpha}}N + b_a) = \frac{\xi_2(b_a)}{2}\varepsilon_\alpha + \mathcal{O}(\varepsilon_\alpha^2). \tag{30}$$

Again, using Lem. 2, we get

$$\mathbb{E}f(Z_{a\alpha})^2 = \eta_{00}(b_a) + (\eta_{11}(b_a) + \eta_{02}(b_a))\varepsilon_\alpha + \mathcal{O}(\varepsilon_\alpha^2). \tag{31}$$

For the covariance calculation, we define $\rho_{\alpha\beta} := c_{\alpha\beta}/\sqrt{c_{\alpha\alpha}}$ and recall that $\rho_{\alpha\beta}$ is order $1/\sqrt{m}$. Thus,

$$\mathbb{E}f(Z_{a\alpha})f(Z_{a\beta}) = \mathbb{E}f(\sigma_W\sqrt{c_{\alpha\alpha}}N' + b_a)f\left(\sigma_W\rho_{\alpha\beta}N' + \sigma_W\sqrt{c_{\beta\beta} - \rho_{\alpha\beta}^2}N + b_a\right) \tag{32}$$

$$= \left(\mathbb{E}_{N'}f(\sigma_W\sqrt{c_{\alpha\alpha}}N' + b_a)\right)\left(\mathbb{E}_N f\left(\sigma_W\sqrt{c_{\beta\beta} - \rho_{\alpha\beta}^2}N + b_a\right)\right) \tag{33}$$

$$+ \left(\mathbb{E}_{N'}\sigma_W\rho_{\alpha\beta}N'f(\sigma_W\sqrt{c_{\alpha\alpha}}N' + b_a)\right)\left(\mathbb{E}_N f'\left(\sigma_W\sqrt{c_{\beta\beta} - \rho_{\alpha\beta}^2}N + b_a\right)\right). \tag{34}$$

For the first term, we get

$$\xi_0(b_a)^2 + \xi_0(b_a)\xi_2(b_a)\frac{\varepsilon_\alpha + \varepsilon_\beta}{2} + \mathcal{O}(m^{-1}) = \mathcal{O}(m^{-1}). \tag{35}$$

For the second term, we have

$$\tilde{c}_{\alpha\beta}\left(1 - \frac{\varepsilon_\alpha + \varepsilon_\beta}{2} + \mathcal{O}(m^{-1})\right)\left(\xi_1(b_a) + \frac{\xi_1(b_a) + \xi_3(b_a)}{2}\varepsilon_\alpha + \mathcal{O}(m^{-1})\right) \tag{36}$$

$$\times\left(\xi_1(b_a) + \frac{\xi_1(b_a) + \xi_3(b_a)}{2}\varepsilon_\beta + \mathcal{O}(m^{-1})\right) \tag{37}$$

$$= \gamma_{1,1}(b_a)\tilde{c}_{\alpha\beta} + \mathcal{O}(m^{-1}). \tag{38}$$

Thus, to second order we have

$$\mathbb{E}f(Z_{a\alpha})f(Z_{a\beta}) = \gamma_{1,1}(b_a)\tilde{c}_{\alpha\beta} + \mathcal{O}(n_0^{-1}). \tag{39}$$

$\square$

## A.1 Linearized model

Define

$$F^{\text{lin}} := \mathcal{C}\Theta^1\Sigma + \left(\mathcal{V}^2 - \mathcal{C}^2\right)^{1/2}\Theta^2, \tag{40}$$

where: $\Theta^1$ and $\Theta^2$ are $n_1 \times m$ matrices that have iid Gaussian entries with mean 0 and variance $1/n_1$; 2) $\mathcal{V}$ and $\mathcal{C}$ are $n_1 \times n_1$ diagonal matrices with entries

$$\mathcal{V}_a^2 = \eta_{00}(b_a) \equiv \eta(b_a) \quad \text{and} \quad \mathcal{C}_a^2 = \gamma_{11}(b_a) \equiv \zeta(b_a), \tag{41}$$

for a bias vector $b$; 3) $\Sigma$ is a matrix square root of $\frac{1}{n_0}X^tX$.

A simple calculation shows that the entries of the matrix $F^{\text{lin}}$ match the first and second mixed moments of $F$ given in Lem. 1. However, unlike $F$, $F^{\text{lin}}$ is linear in the random matrices $\Theta^1$ and $\Theta^2$; in this sense, it is a linearization of $F$.

# B Derivation of self-consistent equation for $F^{\text{LIN}}$

**Theorem 2.** *Let $m_m(z)$ be the Stieltjes transform of $\frac{1}{n_1}F^{lin^t}F^{lin}$, i.e. $\frac{1}{m}\text{tr}G(z)$. Then with probability 1, as $n_1 \to \infty$, for all $z$ such that $\Im z > 0$, $m_{n_1}(z) \to m(z)$, where $m(z)$ is the solution of the coupled equations*

$$m(z) = \mathbb{E}_{S \sim MP(\phi)}\left[\frac{1}{C_0(z) + SC_1(z)}\right] \quad and \quad \tilde{m}(z) = \mathbb{E}_{S \sim MP(\phi)}\left[\frac{S}{C_0(z) + SC_1(z)}\right] \tag{42}$$

*with*

$$C_0(z) := -z + \mathbb{E}_{B \sim \mu_B}\left[\frac{\eta(B) - \zeta(B)}{D(B)}\right], \quad C_1(z) := \mathbb{E}_{B \sim \mu_B}\left[\frac{\zeta(B)}{D(B)}\right], \tag{43}$$

$$D(B) := 1 + \frac{\psi}{\phi}\left(\zeta(B)\tilde{m}(z) + (\eta(B) - \zeta(B))m(z)\right). \tag{44}$$

We want to study the eigenvalues of $\frac{1}{n_1}F^{\text{lin}^t}F^{\text{lin}}$. Note without loss of generality it is sufficient to consdier diagonal $\Sigma$, since we can diagonalize $\Sigma$ using some orthogonal matrices $O$ and $O'$ as it is the square root of a positive definite matrix. Moreover, these orthogonal matrices, when applied to either $\Theta^1$ or $\Theta^2$ do not change their distributions. Thus, diagonalizing $\Sigma$, so that $\Sigma_{\alpha\alpha} = \sqrt{\lambda_\alpha^X}$, where $\lambda_\alpha^X$ are the eigenvalues of $X^tX/n_0$, results in an equivalent matrix ensemble in distribution (see Silverstein & Bai (1995b) for more detail). With this simplification, $F^{\text{lin}}$ has independent entries given by

$$F_{a\alpha}^{\text{lin}} = \mathcal{C}_a\Theta_{a\alpha}^1\sqrt{\lambda_\alpha^X} + \sqrt{\mathcal{V}_a^2 - \mathcal{C}_a^2}\Theta_{a\alpha}^2. \tag{45}$$

To make the derivation easier, we can partly linearize the problem by studying the matrix

$$H := \begin{bmatrix} -zI & F^{\text{lin}^t} \\ F^{\text{lin}} & -I \end{bmatrix}. \tag{46}$$

By the Schur complement formula, one easily finds

$$m_m(z) := \frac{1}{m}Tr(F^{\text{lin}^t}F^{\text{lin}}/n_1 - zI) = \frac{1}{m}\sum_{\alpha=1}^m G_{\alpha\alpha}(z) \tag{47}$$

$$\text{and} \quad z\tilde{m}_m(z) := \frac{1}{n_1}Tr(F^{\text{lin}}F^{\text{lin}^t}/n_1 - zI) = \frac{1}{n_1}\sum_{a=n_1+1}^{n_1+m} G_{aa}(z), \tag{48}$$

where $G$ is the *inverse* of $H$. Again by the Schur complement formula

$$\frac{1}{G_{\alpha\alpha}} = -z - \sum_{a,b=1}^{n_1} F_{a\alpha}^{\text{lin}}F_{b\alpha}^{\text{lin}}G_{m+a,m+b}^{(\alpha)} \tag{49}$$

$$\text{and} \quad \frac{1}{G_{aa}} = -1 - \sum_{\alpha,\beta=1}^m F_{a\alpha}^{\text{lin}}F_{a\beta}^{\text{lin}}G_{\alpha\beta}^{(a)} \tag{50}$$

for $\alpha \in \{1, \ldots, m\}$ and $a \in \{m+1, \ldots, m+n_1\}$ and $G^{(a)}$ is the inverse of the minor $H^{(a)}$. Since $G(\alpha)$ is independent of $\Theta^1_{1\alpha}, \ldots, \Theta^1_{n1\alpha}$ and $\Theta^2_{1\alpha}, \ldots, \Theta^2_{n1\alpha}$, we see by taking the expectation over these variables that

$$\mathbb{E} \sum_{a,b=1}^{n_1} F^{\mathrm{lin}}_{a\alpha} F^{\mathrm{lin}}_{b\alpha} G^{(\alpha)}_{m+a,m+b} = \frac{1}{n_1} \sum_{a=1}^{n_1} \left( \lambda^X_\alpha \zeta(b_a) + \eta(b_a) - \zeta(b_a) \right) G^{(\alpha)}_{m+a,m+a}. \tag{51}$$

Moreover, standard concentration inequalities and the Ward identity (see Erdos & Yau (2017)) show

$$\left| \sum_{a,b=1}^{n_1} F^{\mathrm{lin}}_{a\alpha} F^{\mathrm{lin}}_{b\alpha} G^{(\alpha)}_{m+a,m+b} - \mathbb{E} \sum_{a,b=1}^{n_1} F^{\mathrm{lin}}_{a\alpha} F^{\mathrm{lin}}_{b\alpha} G^{(\alpha)}_{m+a,m+b} \right| \tag{52}$$

$$\leq C \left( \frac{1}{m^2} \sum_{a,b} \left| G^{(\alpha)}_{m+a,m+b} \right|^2 \right)^{1/2} \leq C \left( \frac{1}{m^2} \sum_a \frac{\Im G^{(\alpha)}_{m+a,m+a}}{\Im z} \right)^{1/2} \leq \mathcal{O}(1/\sqrt{m}) \tag{53}$$

with high probability. Similar bounds are easily obtained for $\sum_{\alpha,\beta=1}^m F^{\mathrm{lin}}_{a\alpha} F^{\mathrm{lin}}_{a\beta} G^{(a)}_{\alpha\beta}$.

We may also replace $G^{(\alpha)}$ with $G$ at the expense of another small error that can be bounded using the Cauchy interlacing theorem: $\left| G^{(\alpha)}_{m+a,m+a} - G_{m+a,m+a} \right| \leq \mathcal{O}(1/m)$. Using this control over these sums, we see

$$\frac{1}{G_{\alpha\alpha}} = -z - \sum_{a,b=1}^{n_1} F^{\mathrm{lin}}_{a\alpha} F^{\mathrm{lin}}_{b\alpha} G^{(\alpha)}_{ab} \tag{54}$$

$$= -z - \frac{1}{n_1} \sum_{a=1}^{n_1} \left( \lambda^X_\alpha \zeta(b_a) + \eta(b_a) - \zeta(b_a) \right) G_{m+a,m+a} + \mathcal{O}(1/\sqrt{m}) \tag{55}$$

and

$$\frac{1}{G_{aa}} = -1 - \sum_{\alpha,\beta=1}^m F^{\mathrm{lin}}_{a\alpha} F^{\mathrm{lin}}_{a\beta} G^{(a)}_{\alpha\beta} \tag{56}$$

$$= -1 - \frac{1}{n_1} \sum_{\alpha=1}^m \left( \lambda^X_\alpha \zeta(b_a) + \eta(b_a) - \zeta(b_a) \right) G_{\alpha\alpha} + \mathcal{O}(1/\sqrt{m}). \tag{57}$$

Finally, we invert Eqn. (56), multiply by $\lambda^X_\alpha \zeta(b_a) + \eta(b_a) - \zeta(b_a)$, and average over $a$ to find

$$\frac{1}{n_1} \sum_{a=1}^{n_1} \left( \lambda^X_\alpha \zeta(b_a) + \eta(b_a) - \zeta(b_a) \right) G^{(\alpha)}_{m+a,m+a} \tag{58}$$

$$= -\frac{1}{n_1} \sum_{a=1}^{n_1} \frac{\lambda^X_\alpha \zeta(b_a) + \eta(b_a) - \zeta(b_a)}{1 + \frac{\psi}{\phi} \left( \zeta(b_a) \tilde{m}(z) + (\eta(b_a) - \zeta(b_a)) m_m(z) \right) + \mathcal{O}(1/\sqrt{m})} \tag{59}$$

$$= -\frac{1}{n_1} \sum_{a=1}^{n_1} \frac{\lambda^X_\alpha \zeta(b_a) + \eta(b_a) - \zeta(b_a)}{1 + \frac{\psi}{\phi} \left( \zeta(b_a) \tilde{m}(z) + (\eta(b_a) - \zeta(b_a) m_m(z) \right)} + \mathcal{O}(1/\sqrt{m}) \tag{60}$$

$$= -\mathbb{E}_{B \sim \mathcal{N}(0, \sigma_b^2)} \left[ \frac{\lambda^X_\alpha \zeta(B) + \eta(B) - \zeta(B)}{1 + \frac{\psi}{\phi} \left( \zeta(B) \tilde{m}(z) + (\eta(B) - \zeta(B)) m_m(z) \right)} \right] + o(1), \tag{61}$$

where $\tilde{m}_m(z) = \frac{1}{m} \sum_\alpha \lambda^X_\alpha G_{\alpha\alpha}$, where we Taylor expanded in the second step, and where we used our assumption on $B$.

We can now invert Eqn. (54) and average over $\alpha$ to find

$$m_m(z) \to \mathbb{E}_{S \sim \mu_X} \left[ \frac{1}{-z + \mathbb{E}_{B \sim \mathcal{N}(0, \sigma_b^2)} \left[ \frac{S\zeta(B) + \eta(B) - \zeta(B)}{1 + \frac{\psi}{\phi} (\zeta(B) m_\sigma(z) + (\eta(B) - \zeta(B)) m(z))} \right]} \right]. \tag{62}$$

Similarly,

$$\tilde{m}_m(z) \to \mathbb{E}_{S \sim \mu_X} \left[ \frac{S}{-z + \mathbb{E}_{B \sim \mathcal{N}(0, \sigma_b^2)} \left[ \frac{S\zeta(B)+\eta(B)-\zeta(B)}{1+\frac{\psi}{\phi}(\zeta(B)\tilde{m}(z)+(\eta(B)-\zeta(B)m(z)))} \right]} \right]. \tag{63}$$

Note that the integral over $S$ here is an integral over the limiting distribution of the data $\mu_X$, which we assume

$$\frac{1}{m} \sum_\alpha \delta_{\lambda_\alpha^X} \to \mu_X \tag{64}$$

in distribution. In the case of iid Gaussian data, this is exactly given by the Marchenko-Pastur distribution.

## C ADDITIONAL FIGURES

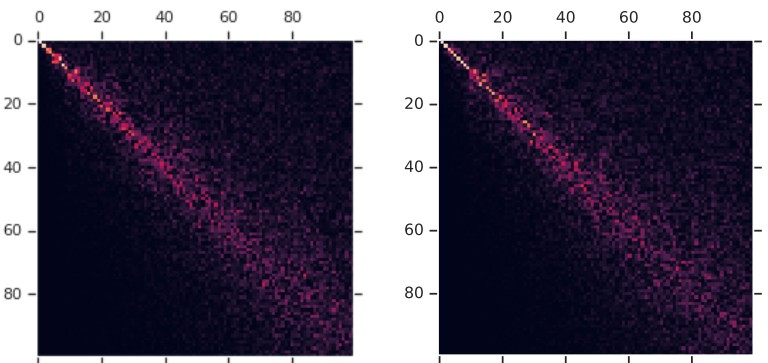

Figure 5: Comparison of right singular vectors of $X$ and $F$ for MNIST (left) and CIFAR10 (right). Entry $ij$ shows $\mathbf{x}_i \cdot \mathbf{f}_j$. Although our theoretical results do not give predictions for how the singular vectors change, we found interesting behavior, with very little change to the largest singular vectors (which are nearly isolated in the spectrum), but more mixing of singular vectors in the dense part of the distribution.

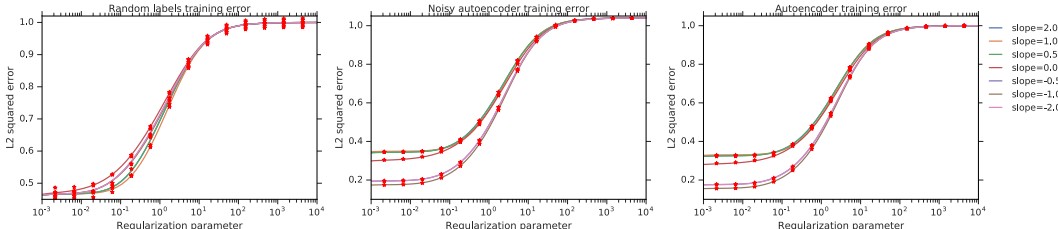

Figure 6: Comparisons of simulated ridge regression error and our theoretical prediction. Here we vary the activation function by changing the slope $\alpha$ in leaky ReLU. In particular $\alpha = -1$ is a linear function, $\alpha = 0$ is regular ReLU, and $\alpha = 1$ is the a scaled absolute value function. We normalize all functions so that $\mathbb{E}_b[\eta(b)] = 1$. We get excellent agreement with theory from only a single sample.

