# OpenReview forum: "A Random Matrix Perspective on Mixtures of Nonlinearities in High Dimensions"
_ICLR.cc/2020/Conference — Reject_

### Official Review · AnonReviewer1 · 2019-10-23
**Official Blind Review #1**

**Rating:** 8

**Review:**

In this work, the authors focus on the high-dimensional regime in which both the dataset size and the number of features tend to infinity. They analyze the performance of a simple regression model trained on the random features and revealed several interesting and important observations. I think it is a solid work and vote for acceptance.
Pros:
(1) This paper has a solid theoretical foundation. Although I have not checked in detail, I think the deduction is clear and the contribution is well-established.
(2) It extends some traditional bounds to more general cases. I think it will provide useful guidance to real applications, such as the network design in deep learning.
(3) The authors have explained the results in a clear way. Thus, it will benefit the following readers and give deep insights about the related research areas.
Minor comments:
(1) I think some assumptions should be explained. For example, why the authors focus only on linear model. Due to the simplicity or the requirement from real applications?
(2) More experimental results on large data sets should be added to validate the effectiveness.

**Experience Assessment:**

I have read many papers in this area.

**Review Assessment: Checking Correctness Of Derivations And Theory:**

I assessed the sensibility of the derivations and theory.

**Review Assessment: Checking Correctness Of Experiments:**

I assessed the sensibility of the experiments.

**Review Assessment: Thoroughness In Paper Reading:**

I read the paper at least twice and used my best judgement in assessing the paper.

---

> ### Author Response · Authors · 2019-11-11
> **Response to minor comments**
>
> (1) We focused on the linear model because it is the simplest (nontrivial) learning task. Moreover, the analysis follows directly from the paper’s main theorem, whereas more complex learning tasks require substantially more calculation—generally beyond the scope of this initial work.
>
> (2) We are happy to include experiments on additional datasets in a final version of the paper.

---

### Official Review · AnonReviewer2 · 2019-10-23
**Official Blind Review #2**

**Rating:** 3

**Review:**

This paper investigates the asymptotic spectral density of a random feature model F(Wx + B).  This is an extension of existing result that analyzed a model without the bias term, i.e., F(WX). This extension requires a modification of the proof technique. In addition to that, it analyzed a mixture of linear and non-linear activation functions, and show that mixture is better than single nonlinearity in terms of expected training error for ridge regression estimators.

Pros:
- This paper investigates an interesting problem and it successfully extends the existing work. The theoretical curve well matches the simulated curve.
- The finding that mixture of nonlinearities gives better expected training error is interesting.

Cons:
- The extension to the model with bias seems a bit incremental. In practice, we may consider an input with additional constant feature, X <- [X,1], to deal with both models in a unified manner. There should be more discussion about why this kind of trivial argument cannot be applied in the analysis.
- The effect of mixture of activation functions is investigated in the "training error," but I don't see much significance on investigating the training error thoroughly. Instead, people are interested in the test error. I guess there does not appear such a trade-off for the test error and the linear activation function would be always better because the true function is the linear model. Hence, more expositions about why the training error is investigated should be provided.

More minor comment:
- I guess the definition of Etrain  (Eq.(17)) requires an expectation with respect to the training data.
- Assumptions of the activation function f should be provided; is it just assumed to be differentiable?, ReLU is included?
- The definition of G(\gamma) in page 6 had better to be consistent to that in previous pages.



**Experience Assessment:**

I do not know much about this area.

**Review Assessment: Checking Correctness Of Derivations And Theory:**

I did not assess the derivations or theory.

**Review Assessment: Checking Correctness Of Experiments:**

I assessed the sensibility of the experiments.

**Review Assessment: Thoroughness In Paper Reading:**

I read the paper at least twice and used my best judgement in assessing the paper.

---

> ### Author Response · Authors · 2019-11-11
> **Response to cons.**
>
> 1. This is a natural question and one we want to respond to in detail. Augmenting X as [X, 1] does not allow us to address both models in a unified manner. In fact, if it did, our theorem would be a special case of earlier work -- in particular, the final spectral distribution would depend on only the two parameters \eta and \zeta. However, the class of spectral distributions we find is nonparametric and thus cannot be expressed in this way.
>
> The issue is that the previous work only applies if the augmented coordinate [--, 1] is transformed by weights of the same order of magnitude as the other features in X, i.e. the bias is of order O(1/\sqrt{n}) per output feature -- an effect that disappears in the large m limit. Instead, our formulation f(WX+b) allows for an O(1) bias term per output feature. One can certainly reformulate this algebraically as f( [W, b] . [X, 1] ), but the scaling assumptions of prior work are (significantly) broken since W and b are on different scales. Addressing these issues is a main contribution of our derivation.
>
> We have included a brief discussion of this point in the new version of the paper.
>
> 2. In our setting the training error is a way to quantify the capacity of the function class. Indeed the test error is also interesting, but analyzing it requires specifying a model for the joint distribution between the data points and labels, and substantially more analysis. Unfortunately, this analysis is outside of the scope of this paper.

---

> ### Author Response · Authors · 2019-11-11
> **Response to minor comments**
>
> 1. The formula for E_train in Eq. (17) is exact, since the training error is a random variable that converges in probability to a deterministic quantity. We have updated the statement of Cor. 1 to clarify this.
>
> 2. The only assumption on f is that it leads to finite moments up to third order under Gaussian expectation. There is also some discussion of the differentiability of f in Remark 4: while we assume f is differentiable for convenience, it is in fact not a necessary assumption as the derivative is only used in a Gaussian expectation, which inherently smooths the function. This implies in particular that ReLU activations are included in the behavior we identify. See Figs. 1 and 2, for example, where we find excellent agreement when f is ReLU.
>
> 3. The object G(\gamma) is indeed the same as G defined in Eq. (1); the only difference is that the function is evaluated at z=-\gamma.

---

### Official Review · AnonReviewer3 · 2019-10-31
**Official Blind Review #3**

**Rating:** 6

**Review:**

This paper analyzed the asymptotic training error of a simple regression model trained on the random features for a noisy autoencoding task and proved that a mixture of nonlinearities can outperform the best single nonlinearity on such tasks.

Comments:
1.The paper is well written and provides sound derivation for the theories.

2. Since this area is out of my expertise, I’m not sure whether merely extending the work of Pennington & Worah (2017) to non-Gaussian data distributions is significant enough or not.

3. Except for Fig 4, the other figures seem out of the context. There is no explanation for the purpose of those figures in the main contents. It is a bit hard for the audience to figure out what to look at in the figures or what the figures try to prove.

4. In “..., and our analysis actually extends to general such distributions, ... ”, “general” should be “generalize”.

5. In “And whether these products generate a medical diagnosis or a navigation decision or some other important output, ..”, “whether” should be “no matter”.

6. “..., they may not be large in comparison to the number of constraints they are designed asked satisfy.” should be “...  they are designed to satisfy”.


**Experience Assessment:**

I do not know much about this area.

**Review Assessment: Checking Correctness Of Derivations And Theory:**

I assessed the sensibility of the derivations and theory.

**Review Assessment: Checking Correctness Of Experiments:**

I assessed the sensibility of the experiments.

**Review Assessment: Thoroughness In Paper Reading:**

I made a quick assessment of this paper.

---

> ### Author Response · Authors · 2019-11-11
> **Response to comments**
>
> 2. As we outline in Subsection 1.1, our work generalizes substantially over previous work as we analyze nontrivial data distributions and NNs with biases. In particular, previous formulae for the spectrum would agree poorly on real datasets like MNIST and CIFAR, whereas we get good agreement as evidenced in Fig. 2. Moreover, to achieve these generalizations, we had to introduce a new method of proof rather than the moment method employed in Pennington & Worah (2017).
>
> 3. While Fig. 4 is indeed important and illustrates a main conclusion of our paper about mixtures of nonlinearities, the other figures are also significant as they demonstrate the validity of the mathematical machinery we use to predict the spectrum on complex datasets. To help provide context on these other figures, we have reworded their captions and made sure that each figure is referenced in the main text.
>
> 4, 5, and 6. We have rewritten these sentence to clarify our meaning and fix any grammatical errors.

---

### Author Response · Authors · 2019-11-11
**Thanks to the reviewers and AC**

We are grateful to all reviewers for their constructive feedback and for the time they took to review our work. We have uploaded a new version of the paper and added detailed responses to their comments below. In particular, we address some technical questions about our paper, and explain the overall merits of our work. We hope this encourages reviewers 2 and 3 to reconsider their scores.

---

### Decision · Program_Chairs · 2019-12-19

**Decision:**

Reject

**Comment:**

In this work, the authors focus on the high-dimensional regime in which both the dataset size and the number of features tend to infinity. They analyze the performance of a simple regression model trained on the random features and revealed several interesting and important observations.

Unfortunately, the reviewers could not reach a consensus as to whether this paper had sufficient novelty to merit acceptance at this time. Incorporating their feedback would move the paper closer towards the acceptance threshold.